# Human Milk Archaea Associated with Neonatal Gut Colonization and Its Co-Occurrence with Bacteria

**DOI:** 10.3390/microorganisms13010085

**Published:** 2025-01-04

**Authors:** Maricarmen Salas-López, Juan Manuel Vélez-Ixta, Diana Laura Rojas-Guerrero, Alberto Piña-Escobedo, José Manuel Hernández-Hernández, Martín Noé Rangel-Calvillo, Claudia Pérez-Cruz, Karina Corona-Cervantes, Carmen Josefina Juárez-Castelán, Jaime García-Mena

**Affiliations:** 1Departamento de Genética y Biología Molecular, Cinvestav, Av. Instituto Politécnico Nacional 2508, Mexico City 07360, Mexico; maricarmen.salas@cinvestav.mx (M.S.-L.); juan.velez@cinvestav.mx (J.M.V.-I.); or dianarogue@hotmail.com (D.L.R.-G.); apinae@cinvestav.mx (A.P.-E.); jose.hernandezh@cinvestav.mx (J.M.H.-H.);; 2Institute of Environmental Sciences, Faculty of Biology, Jagiellonian University, Gronostajowa 7, 31-007 Kraków, Poland; 3Hospital General “Dr. José María Rodríguez”, Ecatepec de Morelos 55200, Mexico; drrangelcalvillo@gmail.com; 4Departamento de Farmacología, Cinvestav, Av. Instituto Politécnico Nacional 2508, Mexico City 07360, Mexico; cperezc@cinvestav.mx; 5Institute for Obesity Research, Monterrey Institute of Technology and Higher Education, Monterrey 64849, Mexico

**Keywords:** breastfeeding, human milk, microbiota, Archaea, neonatal gut, 16S rDNA, vertical transmission

## Abstract

Archaea have been identified as early colonizers of the human intestine, appearing from the first days of life. It is hypothesized that the origin of many of these archaea is through vertical transmission during breastfeeding. In this study, we aimed to characterize the archaeal composition in samples of mother-neonate pairs to observe the potential vertical transmission. We performed a cross-sectional study characterizing the archaeal diversity of 40 human colostrum-neonatal stool samples by next-generation sequencing of V5–V6 16S rDNA libraries. Intra- and inter-sample analyses were carried out to describe the Archaeal diversity in each sample type. Human colostrum and neonatal stools presented similar core microbiota, mainly composed of the methanogens *Methanoculleus* and *Methanosarcina*. Beta diversity and metabolic prediction results suggest homogeneity between sample types. Further, the co-occurrence network analysis showed associations between Archaea and Bacteria, which might be relevant for these organisms’ presence in the human milk and neonatal stool ecosystems. According to relative abundance proportions, beta diversity, and co-occurrence analyses, the similarities found imply that there is vertical transmission of archaea through breastfeeding. Nonetheless, differential abundances between the sample types suggest other relevant sources for colonizing archaea to the neonatal gut.

## 1. Introduction

Human milk is composed of essential nutrients and bioactive constituents, including proteins, carbohydrates, fatty acids [1], cytokines [2], oligosaccharides, immune factors [3], and microbiota [4] that cater to the evolving needs of the growing infant [5]. Human milk can be classified into colostrum, transitional, and mature milk according to the composition and lactation stage [6]. For instance, colostrum is the first milk produced during the first days after birth [7]. It is characterized by a high content of growth factors and immunoglobulins, which provide passive immunity to newborns, protecting them against infections [8].

Human milk’s microbiota includes bacteria, eukaryotes, fungi, and archaea [9]. Most studies have focused on the former [10], thus revealing the central bacteriome of human milk and its impact on newborn health [11]. The origin of bacteria in human milk is not fully understood, and two non-exclusive sources have been proposed: the entero-mammary pathway and the retrograde flow [12,13,14]. The first consists of immune cell-mediated bacterial translocation from the maternal gastrointestinal tract to the mammary gland [4,12,13,14]. In more detail, dendritic cells penetrate the gut epithelium and select bacteria, which are then transported to the mammary gland through lymphatic circulation [4,15,16]. The second refers to external contamination or the transfer from the mother’s skin or infant’s mouth to the mammary gland [14,16].

In contrast to bacteria, archaea diversity in human milk has long been neglected [9]. The first evidence of archaea presence in human milk came from metagenomics studies [9,17]. Nevertheless, a recent study proved archaea viability by cultivating *Methanobrevibacter smithii*, a methanogenic archaeon, from colostrum and mature milk [18]. Interestingly, the presence of archaea is reported in the gastrointestinal tract from the first days of life [19,20,21,22]. Accordingly, *M*. *smithii* has been identified as an early colonizer, establishing in the gastric mucosa just after birth [20].

Methanogenic archaea (MA) are prevalent archaea in the digestive tract of adults, particularly *M*. *smithii* and *Methanosphaera stadmanae* [23,24,25,26]. MA plays a fundamental role in the human gut, as they are capable of producing methane (CH_4_) through the assimilation of H_2_ and CO_2_, which are products of polysaccharide fermentation by bacteria [27,28,29,30]. They use hydrogen as an electron donor, reducing CO_2_, acetate, and multiple methyl-containing compounds into CH_4_ [30,31]. This metabolic activity facilitates the growth of fermentative bacteria in the gut, thus conforming to a syntropic relationship [31,32]. MA has been associated with various diseases such as diverticulosis [33,34], inflammatory bowel disease [35,36], atherosclerosis [37,38,39], malnutrition [40], and obesity [32,41,42,43]. However, the relationship between MA and illness is not entirely understood and can be contradictory. For example, MA has been associated with obese and normal-weight individuals [18,41,42,43,44,45,46]. Although the role played by archaea in disease is not clear, they appear to be key microbiota components of the human gastrointestinal tract [47,48].

Given its significance for human health, its presence in human milk, and its role in the gastrointestinal tract from the earliest days of life, this study aimed to characterize the diversity of the archaeal community in human colostrum and neonatal stool samples. This was achieved through Ion Torrent semiconductor DNA sequencing of V5–V6 regions of 16S rRNA gene libraries. Our findings suggest that human milk serves as a primary source of archaea for the neonatal gut, supporting the concept of vertical transmission of archaea during breastfeeding.

## 2. Materials and Methods

### 2.1. Study Design and Selection of Subjects

The cross-sectional descriptive study consisted of 40 mother-neonatal pairs of patients from the “Dr. José María Rodríguez” General Hospital, located in Ecatepec-de-Morelos, State of Mexico (19°36′35″ N, 99°3′36″ W). The samples were obtained from healthy lactating women and exclusively breastfed newborns. Colostrum and neonatal stool samples were collected from 0 to 3 days after birth, up to 2 h after the newborn was breastfed, from November 2017 to January 2018. The inclusion criteria were as follows: (1) Mexican origin with at least two generations of ancestry, (2) gestational age between 37 and 41 weeks, (3) birth weight between 2500–4500 g, (4) Apgar score greater than 7 at 5 min after birth. Exclusion criteria: (1) Maternal probiotic and alcohol consumption, (2) smoking, (3) diabetes before or during pregnancy, (4) antibiotic use during the last trimester of pregnancy and before sampling. The participants were given a survey recording sociodemographic and clinical data (maternal age, gestational age at delivery, type of delivery, newborn sex, and age). Written informed consent was obtained from all participants before the study, following the 2013 Declaration of Helsinki. The protocol was approved by the Ethics Committee of the General Hospital “Dr José María Rodríguez” (Project identification code: 217B560002018006).

### 2.2. Sample Collection

The colostrum-neonatal stool sample pairs were collected on the same day up to 2 h after the newborn was breastfed. Human colostrum (HC) was collected manually into a sterile polypropylene tube (~5 to 10 mL). Breast sanitation was not conducted in order to obtain the microorganisms transferred during breastfeeding. The neonatal stool (NS) was recovered from diapers and transferred to sterile containers using sterile tongue depressors. The samples were sent to the laboratory in a cold environment, distributed in aliquots of 1 mL (HC) or 200 mg (NS), and stored at −20 °C until DNA was extracted within 24 h of arrival.

### 2.3. DNA Extraction

First, 1 mL of HC was centrifuged at 4 °C, 10,000× *g* for 15 min in a refrigerated centrifuge (Eppendorf 5415R) and the fat was removed with a roll of sterile dental cotton. The aqueous supernatant was removed by decantation, the pellet was resuspended in 1 mL of sterile PBS pH 7.4, and then centrifuged again at 10,000× g for 15 min. The obtained pellet was resuspended in 300 μL of PBS pH 7.4 and processed for DNA extraction using FavorPrep Milk bacterial DNA extraction kit (Cat: FAMBD001, Favorgen, Biotech Corp, Taiwan) following the manufacturer’s instructions. Fecal DNA was extracted from 200 mg NS samples using a QIAamp DNA Stool Mini Kit (Cat.: 12830-50, Qiagen group, Venlo, Netherlands), following the manufacturer’s instructions. In two cases, 300 μL of PBS pH 7.4 was used as a negative control for DNA extraction. The DNA concentration in samples was measured at 260/280 absorbance using a NanoDrop 2000 spectrophotometer (Thermo Scientific, Waltham, MA, USA); no absorbance was detected for negative controls. DNA integrity was assessed by electrophoretic fractionation of 5 μL of DNA sample in 0.5% agarose gel stained with 0.80 μL of Midori Green advanced dye (1:15) using TBE buffer. DNA was visualized using the MolecularImager^®^ Gel DocTM XR System program (Bio-Rad Laboratories, Chicago, IL, USA). Extracted DNA was stored at −20 °C.

### 2.4. Preparation of the 16S rRNA Gene Library and Next-Generation Sequencing

The forward Arc787F (5′-ATT-AgA-TAC-CCG-BgT-AgT-CC-3′) and reverse primer Arc1059R (5′-gCC-ATg-CAC-CWC-CTC-T-3′) were used for the polymerase chain reactions (PCR) [49]. The PCR reactions were performed using the Phusion High-Fidelity PCR Kit (Cat F-530S), ThermoFisher Scientific, Waltham, MA, USA). The reaction mixture consisted of 4.0 µL of 1× HF buffer, 0.4 µL of dNTPs (200 µM), 0.2 µL of Phusion polymerase (0.02 U/µL), 1.0 µL of each Forward and Reverse primer (10 µM), and 0.2 µL of MgCl_2_ (0.5 mM). The DNA template volume was adjusted to 13.2 µL with nuclease-free water for a final concentration of 8.0 ng in 20.0 µL. The reactions were programmed in 2720 Thermal Cycler (Applied Biosystems™, ThermoFisher Scientific, Waltham, MA, USA) with a 5 min 95 °C hot start, followed by 5 min initial denaturation at 95° C, 25× (94 °C, 15 s denaturation, 56 °C, 15 s annealing, 72 °C for 15 s extension), and a final 7 min extension at 72 °C. Archaeal DNA from a bioreactor [50] was used as positive control. Blank reactions (PCR products with no DNA template from the DNA extraction pipeline) were used as negative controls. The 358 bp amplicons were fractionated in 1.5% agarose gel dyed with Midori Green (Nippon Genetics^®^, Dueren, Germany) in 0.5× TBE using GeneRulerTM100 bp DNA Ladder (Cat. 15628019, ThermoFisher Scientific, Waltham, MA, USA). Electrophoresis lasted 45 min at 80 V. The DNA was visualized using the Molecular Imager^®^ Gel DocTM XR System program (Bio-Rad Laboratories, Chicago, IL, USA). The library was purified using 2% E-Gel™ EX stained with SYBR GOLD DNA (Cat. G401002, Thermo Scientific, Waltham, MA, USA). The library size and concentration were assessed with the 2100 Bioanalyzer equipment and High Sensitivity DNA kit (Agilent Technologies, Santa Clara, CA, USA). PCR emulsion was carried out with Ion One Touch™ (Life Technologies, Carlsbad, CA, USA) according to the manufacturer’s instructions. Amplicon enrichment with ionic spheres was carried out using Ion OneTouch ES (Life Technologies, Carlsbad, CA, USA). Sequencing was performed using the Ion 318 Kit V2 Chip (Cat. 4488146, Life Technologies, Carlsbad, CA, USA). Ion Torrent PGM software v4.0.2 was used to demultiplex the sequence data based on their barcodes, reads were filtered to exclude low-quality (quality score ≤ 20), polyclonal sequences (homopolymers > 6), and the adapters were trimmed. The datasets generated for this study can be found in the NCBI BioProject ID PRJNA1018680.

### 2.5. ASV Determination and Taxonomic Annotation

The FASTQ files were further processed and analyzed with QIIME 2022.2 [51]. The ASVs (Amplicon Sequence Variants) were determined with the QIIME dada2 denoise-single plugin, with sequence truncation at 238 nt. The taxonomic assignation was done with the feature-classifier classify-consensus-blast plugin, with a 97% percentage identity. The Greengenes 2 database was used for analysis with BLAST [52].

### 2.6. Bioinformatic Analyses

R 4.2.0 [53] in Rstudio [54] was used for the relative abundance, diversity, differential, discriminatory, and co-occurrence analyses. For that purpose, the following packages were used: to import the qiime artifacts, qiime2R [55], for alpha and beta diversity analyses, phyloseq 1.4.0 [56], for differential analyses, DESEq2 1.3.6 [57] and ALDEx2 1.36.0 [58], for discriminatory analysis, lefser 1.15.9 [59], for the heatmap elaboration, ComplexHeatmap 2.12.0 [60], for the ANOSIM, vegan 2.6-2 [61], and for graphical images, tidyverse 1.3.1 [62], dplyr 1.09 (data frame manipulation), ggplot2 3.3.6, scales 1.2.0, ggpubr 0.4.4, and gridExtra 2.3 [63]. PICRUSt2 was executed with the MetaCyc metabolic pathway database option following a published pipeline tutorial [64]. Co-occurrence network analysis was conducted with microeco [65] and meconetcomp [66] packages using 0.6 Spearman’s rank correlation coefficient with a 99% confidence level. All analyses were filtered to the Archaeal Domain except for the co-occurrence analysis. In addition, the reads were filtered using a stringent pipeline to ensure data quality. First, potential contaminants identified in negative PCR sequencing controls were removed using decontam 1.24.0 [67]. Next, low-abundance ASVs (defined as those with a prevalence below 0.1% based on a Phred quality score < 30) were excluded. Finally, sparse ASVs (those > 90% zero values across samples) were filtered out.

### 2.7. Statistical Methods

Archaeal diversity within samples was estimated with alpha diversity, determining Observed ASVs, Shannon, Simpson, and Fisher indexes. The Shapiro–Wilk preliminary test was applied to test if the data followed a normal distribution; then, the Kruskal–Walli’s test was applied. The differences in beta diversity between samples were evaluated by ANOSIM (Analyses of similarities). A differential abundance analysis (DESeq2) for dependent samples was performed to identify relevant taxa in the distinct sample types and pairs and was evaluated with a Wald test; *p*-values were adjusted with Holm–Bonferroni method statistics. *p-* or *q*-values < 0.05 were considered significant.

## 3. Results

### 3.1. Most Participant Mother-Neonate Pairs Were from Urban Areas

Forty colostrum and neonatal stool samples were collected within four days immediately after birth. Most participants came from the State of Mexico (19.4969° N, 99.7233° W) or Mexico City (19.4326° N, 99.1332° W) low-income areas (Table 1). Almost all mothers work at home (95%), and close to 50% of them have a high school education. Most of them were normal weight and more than half of the deliveries were vaginal (Table 1). Regarding the neonate’s anthropometric data, 60% were females and 35% were males; information was not available for two of the samples.

### 3.2. Methanoculleus and Methanosarcina Lead the Core Archaeal Community in Mother-Neonate Pairs

The archaeal composition and diversity in colostrum and neonatal stool samples from mother-neonate pairs were analyzed using high-throughput sequencing of the V5–V6 regions of the 16S rRNA gene. A total of 4,531,448 raw sequences were generated, including 2,269,715 from colostrum and 2,261,733 from neonatal stool, with a median Phred quality score of 27 (Appendix A, Appendix A).

Archaeal sequence reads were filtered and decontaminated using a stringent pipeline as described in the Section 2.6 (Decontam results are shown in Appendix A). After filtering, two archaeal phyla were identified in both sample types. Halobacteriota was the predominant phylum, accounting for nearly 89% of the archaeal abundance, followed by Methanobacteriota_A_1229 (~11%) (Figure 1A, Appendix A).

At the genus level, the dominance of *Methanoculleus*_A_2118 and *Methanosarcina*_2619 was observed in decreasing order of abundance, followed by *Methanobrevibacter*_A and *Methanothermobacter*_A_884. The least abundant archaeal genus detected was *Methanofollis* (Figure 1B, Appendix A). No differences or trends with statistical significance were observed in archaeal abundance between the two sample types. These findings suggest a consistent core archaeal community composition between colostrum and neonatal stool.

### 3.3. Colostrum Is the Primary Source of Neonatal Archaeal Communities

To understand the origin of archaea in neonatal stool samples, a SourceTracker analysis was performed using the unfiltered, non-decontaminated reads (Figure 2). This analysis estimates the proportion of archaea originating from a specific source—in this case, human colostrum. Sequencing results of negative controls from PCR were also included in the study. Our results indicated that nearly three-quarters of the archaeal communities present in the neonatal stool were derived from colostrum. At the same time, the remainder originated from an unidentified source; no trace of contamination from negative controls was observed (Figure 2A).

Concerning specific genera, *Methanosarcina*_2619, *Methanoculleus*_A_2118, *Methanothermobacter*_A_884, and *Methanobrevibacter*_A were found to originate predominantly from colostrum. However, portions of *Methanobrevibacter*_A, *Methanoculleus*_A_2118, *Methanosarcina*_2619, and *Methanothermobacter*_A_884 ASVs could not be traced back to colostrum or negative controls. In addition, some archaea, such as *Methanobacterium*_A_1053 and different *Methanosarcina* and *Methanofollis*, had unknown origins. Also, several less-known archaeal genera, including *Nitrosocosmicus*, LDKUD01, RumEn-M2, TA-21, UBA10452, VadinCA11, and various unassigned taxa, had origins that could not be explained by colostrum or any other analyzed source (Figure 2B).

### 3.4. Colostrum and Neonatal Stool Exhibited Comparable Diversity Metrics

Alpha and beta diversity analyses were conducted to evaluate the homogeneity within and among samples. For the alpha diversity, an interesting trend emerged: colostrum appeared more diverse than neonatal stool across all indices, albeit with a small effect size (Figure 3A). The exception was the Inverse Simpson index, which showed a medium effect size. Statistical tests, including Kruskal–Wallis and ANOVA, revealed significant differences for all indices except the observed number of ASVs. However, after applying Holm correction to adjust the *p*-values, none of the differences remained statistically significant (Figure 3A, Appendix A).

Beta diversity was analyzed using weighted UniFrac distance and Jensen–Shannon divergence. It is important to note that filtering the reads resulted in some samples having zero features. To avoid bias in hypothesis testing, complete mother-neonate pairs were retained, including those with zero-feature samples. Our results indicate no significant differences between colostrum and neonatal stool communities, as evidenced by the single cluster formed in Figure 3B,C and the lack of statistical significance.

However, considerable distances were observed between paired samples (mother-neonate binomials), which none of the measured variables explain. These distances are represented by the dotted lines connecting each pair reflected with the low R-values from the ANOSIM analysis (−0.001 for Jensen–Shannon divergence and −0.02 for weighted UniFrac). This variation is likely attributed to the inclusion of zero-feature samples in the analysis (Figure 3B,C).

### 3.5. There Was a Moderate Positive Correlation in Methanoculleus_A_2118 Abundance in the Binomial

Differential abundance analyses using DESeq2, ALDEx2, and LEFSE did not reveal any significant differences. The two-group Pearson correlation test was used to detect potential similarities. A moderate positive correlation (*ρ* = 0.35, *p* = 0.038) for *Methanoculleus*_A_2118 was found for each pair in the binomials (Figure 4A). However, after applying the Holm correction, this correlation was no longer statistically significant. Correlations between ASVs and variables such as mothers’ age, pre-pregnancy BMI, and weight gain using the ALDEx2 correlation module were also examined, but no associations were found. The results for *Methanoculleus*_A_2118 are shown in Figure 4B–D.

### 3.6. Co-Occurrence Networks Show Distinct Microbial Relationships in Colostrum and Neonatal Stool

The co-occurrence networks of microbial interactions in each sample type were analyzed. Since the sequencing data included bacterial reads, bacteria were incorporated into the analysis to explore their relationships with archaea. Separate networks were constructed for colostrum and neonatal stool using Pearson correlation with *p* < 0.01 and *ρ* > 0.6 thresholds. The colostrum network contained 97 unique edges, while the neonatal stool network had 77, with 133 edges (43.3%) shared between the two (Figure 5A).

The most notable relationships are highlighted in Figure 5B. Many involved ASVs from the same genus (e.g., *Streptococcus*-*Streptococcus*, *Methanoculleus*-*Methanoculleus*), which may suggest a phylogenetic similarity. However, relationships between different genera in the same phylogenetics domain, such as *Staphylococcus*-*Streptococcus*, *Methanoculleus*-*Methanosarcina*, *Corynebacterium*-*Staphylococcus*, *Corynebacterium*-*Streptococcus*, and *Bifidobacterium*-*Streptococcus* were observed. Remarkably, a direct inter-domain relationship of bacteria with archaea was limited only to *Collinsella*-*Methanosarcina* and *Methanoculleus*-*Streptococcus* (Figure 5B).

Examining the overall topology of the individual networks for neonatal stool (Figure 5C, Appendix A) and colostrum (Figure 5D, Appendix A), most edges were between members of the same genus or domain, with only a few hubs showing archaea-bacteria relationships. Notably, bacterial hubs frequently served as bridges for distant connections. Additionally, the hubs in neonatal stool networks were more dispersed, while those in colostrum networks were more closely clustered.

### 3.7. Predicted Functional Metagenome Highlights Methanogen-Associated Pathways

Finally, functional metabolic pathways in the colostrum and neonatal stool microbiota were identified using PICRUSt2 analysis based on the ASV table. No differentially abundant pathways were detected between the two sample types. The primary pathways, present in at least 10% of the samples, were associated with methanogens. These included the incomplete reductive TCA cycle pathway and the L-isoleucine biosynthesis IV and II pathways. Additionally, some pathways were directly related to methanogenesis, such as the factor 420 biosynthesis pathway and the methanogenesis from H₂ to CO₂ pathway (Figure 6).

## 4. Discussion

This work reports the archaeal composition found in colostrum and neonatal stool collected from Mexican mother-neonate pairs. To our knowledge, this is the first report in Mexican binomials. In general, we found a high abundance of phylum Halobacteriota (~89%) and phylum Methanobacteriota_A_1229 (~11%) in colostrum and neonatal stool samples. The main genera were members of this phylum (*Methanoculleus* and *Methanosarcina*). A member of the *Methanoculleus* genus had already been found in the human intestinal mucosa by sequence analysis of the archaeal methyl coenzyme-M reductase (*mcr*A) gene in colonic biopsies [68]. However, apart from this study, little is known about its presence in the human gut. Similarly, in the case of the *Methanosarcina* genus, there are no direct reports [39,69]. Although recent studies have not found this genus [25,26], there is one report on human gut methanogens where a *mcr*A gene phylotype, named Mx-01, has been reported, which was later associated with the Methanosarcinales order [70,71,72,73]. Nevertheless, this phylotype was also suggested to belong to a new order of methanogens [71]. However, it is important to mention that *Methanosarcina* has been found in the gastrointestinal tract of different animals like goats, termites, and pigs [74,75,76]. Alternatively, *Methanosarcina* spp. is part of humans’ oral and skin microbiota [77,78,79,80,81,82,83]. Particularly, the species *Methanosarcina mazeii* and *Methanosarcina vacuolata* have been observed in subjects affected by periodontitis and healthy subjects [78,82]. It could be the case that the presence of this genus in the neonate’s gut was due to oral or skin microbiota. Another hypothesis would be that it has not been well characterized in the human intestine, its abundance being underestimated in infant populations.

*Methanobrevibacter smithii* and *Methanosphaera stadmanae* are considered the most prevalent archaea in the adult human gut [23,25,84]. In addition, in neonates’ gastrointestinal tract, *Methanobrevibacter* and *Methanosphaera*, as well as one uncultured phylotype, have been reported [19,20,21,22]. These studies consisted of 16S rRNA gene sequencing [22], multispacer sequence typing [20], clone library sequencing [19], and qPCR targeting [21]. In our work, the genus *Methanobrevibacter*_A was present in the two sample types but at low abundance (~6.6%), the genus *Methanothermobacter*_A_884 at ~5.5%, and the genus *Methanofollis* in less than 0.1% abundance. The disparities in the archaeal genera proportions of this work against previously reported research could be attributed to lifestyle differences, since no previous reports of archaeal composition in Mexican women exist. It is known that hormonal changes during pregnancy affect the microbiota [85,86,87]. Therefore, the archaeal population of a pregnant woman might be distinct. Moreover, the microbiota is also influenced by diet [88,89,90,91], geographical location, and urban or rural lifestyle [92,93]. Considering this, we hypothesized that the differences found in the proportions of archaeal genera might be because we studied human milk and neonatal stool samples of less than four days of age.

The alpha diversity analysis showed that human colostrum had a tendency for higher archaeal diversity than neonatal stools. Accordingly, human milk bacteria in a similar Mexican cohort also showed higher diversity when compared to neonatal stool [94]. We believe this is due to the neonate’s age (<4 days), which indicates that colonization of their gastrointestinal tract was just beginning, explaining the lower diversity [19,22,70]. Apart from this, the significant difference between colostrum’s and neonate stool’s diversity can also be explained by their ecological niches, which might favor the presence of specific archaea [95]. Interestingly, some authors have concluded that alpha diversity measures might underestimate microbiota, and more robust statistical methods might be necessary to assess the differences [96,97]. Colostrum and neonatal stool were found to have a highly similar core archaeal microbiota, consisting of members of Halobacteriota and Methanobacteriota_a_1229 phyla, which included the genera *Methanoculleus*_A_2118, *Methanosarcina*_2619, *Methanobrevibacter*_A, *Methanothermobacter*_A_884, and *Methanofollis*. The beta diversity analysis, both by NMDS ordination and Jensen–Shannon diverge (JSD) distances, further explained the similarity between colostrum and neonatal stool samples, with the NMDS ordination showing that the sample types overlapped. The similarity between the two sample types suggests that shared taxa are possibly transmitted via breastfeeding. Moreover, the metagenomic prediction analysis suggested no differentially abundant pathways between the sample types, strengthening our previous results.

Regarding the predicted metabolic routes detected, the low abundance could be due to the combination of the natural low abundance of archaea in the sample types and the lack of archaeal metabolic information in the MetaCyc database. Observing the most prevalent pathways among the samples, we found that all of them were associated with methanogens. The incomplete reductive TCA pathway was present in almost 90% of the samples. This route is characteristic of methanogens and allows for the synthesis of intermediates needed for amino acid production [98,99]. The methanogenesis from H_2_ and CO_2_ was also prevalent although less abundant, thus revealing the predominance of hydrogenotrophic archaea, i.e., methanogens that utilize H_2_, formate, or simple alcohols as electron donors and CO_2_ as an electron acceptor [100]. This pathway starts with CO_2_ activation and is followed by numerous transformations, including one aided by the factor 420, which was also prevalent in our study samples, and another methanogenesis indicator [101]. Finally, the L-isoleucine biosynthesis pathways II and IV have been associated with methanogens such as *Methanocaldococcus jannaschii* [102], *Methanothermobacter thermautotrophicus*, and *Methanobrevibacter arboriphilus* [103]. Together, these results suggest that methanogens are vertically transmitted through lactation.

Lastly, the co-occurrence network revealed there were 133 edges or connections (accounting for 43.3%) in common between the human colostrum and neonatal stool networks, supporting the idea of microbial similarity among the sample types. Methanogens and bacteria are known to form syntrophic interactions [24,46,83,104], therefore we sought to see possible Archaea-Bacteria associations. We observed a direct inter-domain relationship of bacteria with archaea only for *Collinsella*-*Methanosarcina* and *Methanoculleus*-*Streptococcus*. Thus, an interaction between these bacteria and the archaea is plausible. In the human colostrum network, we observed *Streptococcus* associated with methanogens. *Streptococcus* co-occurrence with archaea had only been seen in febrile patients’ blood [46,105]. These results showed that some associations found in pathogenesis might also be common in healthy individuals. In the neonatal stool network, we found co-occurring *Methanoculleus*-*Streptococcus*.

The presence of archaea in biological samples like human milk could be questioned since these microorganisms are reported to inhabit extremophile environments. It could be argued that the detection of these species is due to contamination during sample handling or other sources like DNA extraction kits [106]. This last possibility was ruled out in this work by using appropriate handling of samples, negative controls for PCR, and massive DNA sequencing followed by downstream bioinformatic strategies used for removing contaminants and low prevalence features, as described previously in the text. A more interesting explanation is that methanogenic archaea are vertically transmitted by the mother to the neonate, favoring the remotion of hydrogen [107]. It is known that under some conditions, the accumulation of hydrogen in the gut has been reported to be associated with human disease [104,108]. Finally, live archaea have been detected and isolated from human colostrum and milk [18,109].

## 5. Conclusions

In summary, in this study, we characterized the archaeal composition of human colostrum-neonatal stool paired samples, finding evidence of the presence of *Methanoculleus_A_2118*, *Methanosarcina_2619*, *Methanobrevibacter_A*, *Methanothermobacter_A_884*, and *Methanofollis* to be the main genera in both sample types. Moreover, the similarities between the sample types suggest there is vertical transmission of archaea during breastfeeding. Differential abundance yielded no significant differences. Finally, the co-occurrence network analysis showed associations between Archaea and Bacteria that might be relevant for these organisms’ presence in the human milk and neonatal stool ecosystems. Future studies should aim to characterize other potential sources of archaea in the neonatal stool as well as their associations with Bacteria. All in all, this study represents a first step in understanding the origin of archaea in the gut from the beginning of life and remarks on the importance of continuing to study these often-overlooked microorganisms.

## Figures and Tables

**Figure 1 microorganisms-13-00085-f001:**
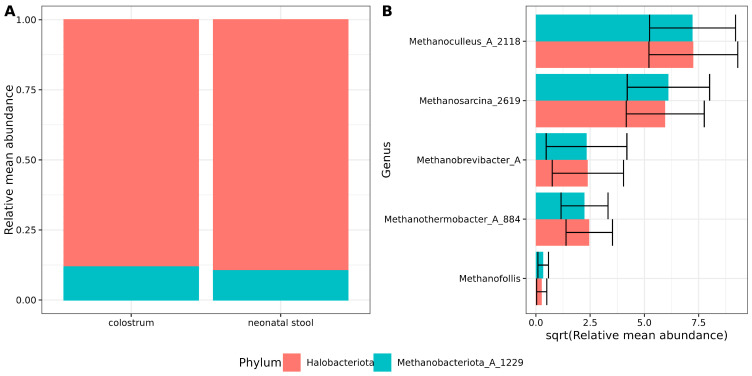
Archaeal composition in samples. (**A**) Stacked bar plots showing the archaeal relative mean abundance at the phylum level. Red and green colors indicate sample type and relative abundances are shown on the *Y*-axis. (**B**) Bar plots showing the square root relative mean abundance of archaea genera. The *Y*-axis labels indicate genus, while the *X*-axis shows the square root-relative and mean abundance. Error bars indicate the standard error of the mean.

**Figure 2 microorganisms-13-00085-f002:**
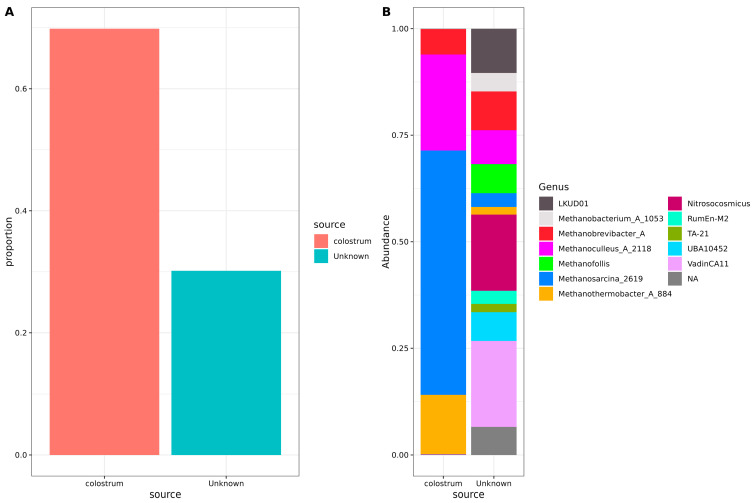
Source tracker analysis of possible archaeal origin in neonatal stool samples. (**A**) Bar plots indicate the total source proportion. The *Y*-axis shows the proportion of the source for the neonatal stool samples, and the *X*-axis shows the source (colostrum or unknown). (**B**) Stacked bar plots showing genera composition for each source. The *Y*-axis shows the genera abundance, and the *X*-axis shows the source (colostrum or unknown). Color sectors indicate the source or the archaeal genus, according to the labels beside the graphics.

**Figure 3 microorganisms-13-00085-f003:**
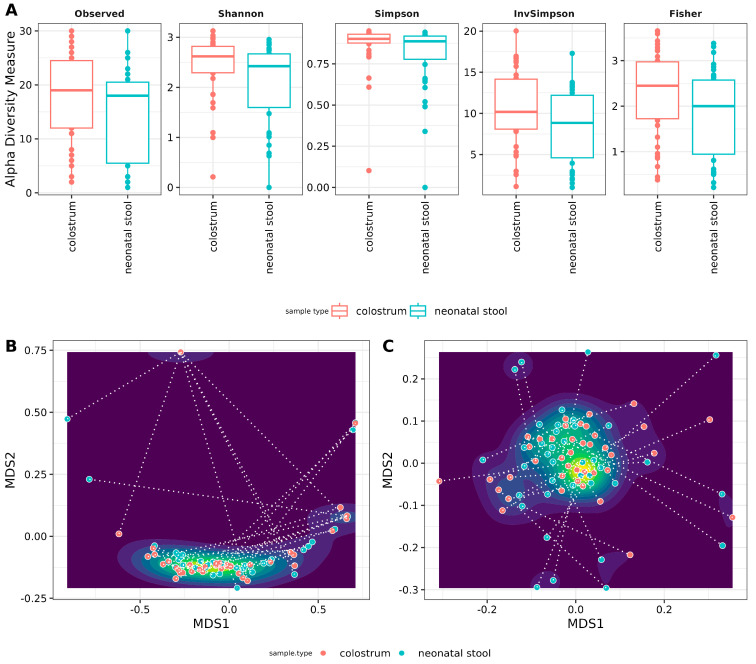
Archaeal diversity in samples. (**A**) Alpha diversity box plots. The *Y*-axis indicates values for the species richness (Observed), diversity indexes (Shannon, Simpson, and Inverse Simpson), and evenness (Fisher). The sample type is shown at the *X*-axis (see Appendix A, for numerical data of indexes) Beta diversity Non-Metric Multidimensional Scaling (NMDS) scatter plots. The graphics show archaeal beta diversity calculated by (**B**) NMDS ordination based on weighted UniFrac and (**C**) Jensen–Shannon diverge (JSD) distances. Sample types (colostrum and neonatal stool) are similar according to ANOSIM (weighted UniFrac R = −0.02, *p* = 0.933 and JSD: R = −0.001 *p* = 0.494).

**Figure 4 microorganisms-13-00085-f004:**
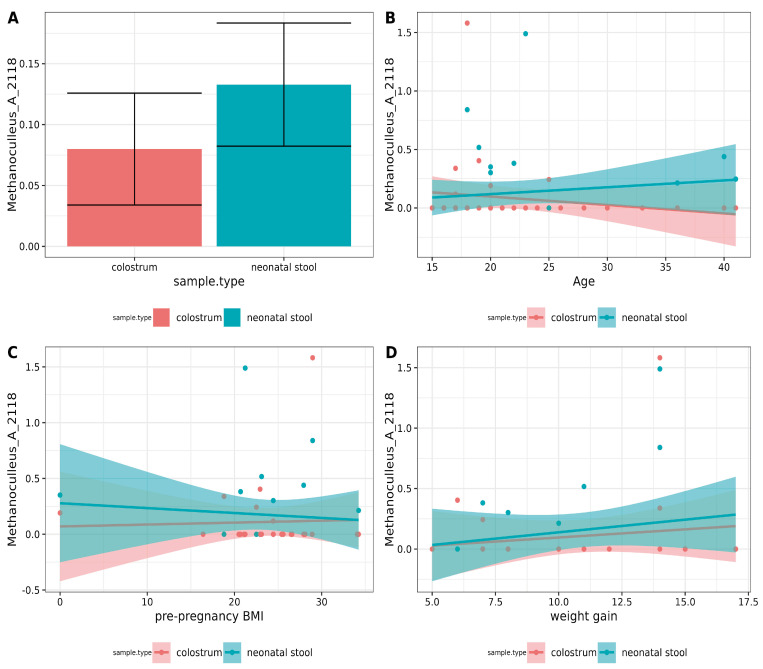
Correlation analysis of archaea between pairs and metadata. (**A**) Paired Pearson correlation test between colostrum and neonatal stool. Only *p*-values < 0.05 were included. The *Y*-axis shows the square root of mean relative abundance, and the *X*-axis, sample types. (**B**–**D**). Correlation analysis between archaea and metadata using aldex.cor module. No significant correlation was found with the mother’s (**A**) age, (**B**) pre-pregnancy BMI, or (**D**) weight gain during pregnancy.

**Figure 5 microorganisms-13-00085-f005:**
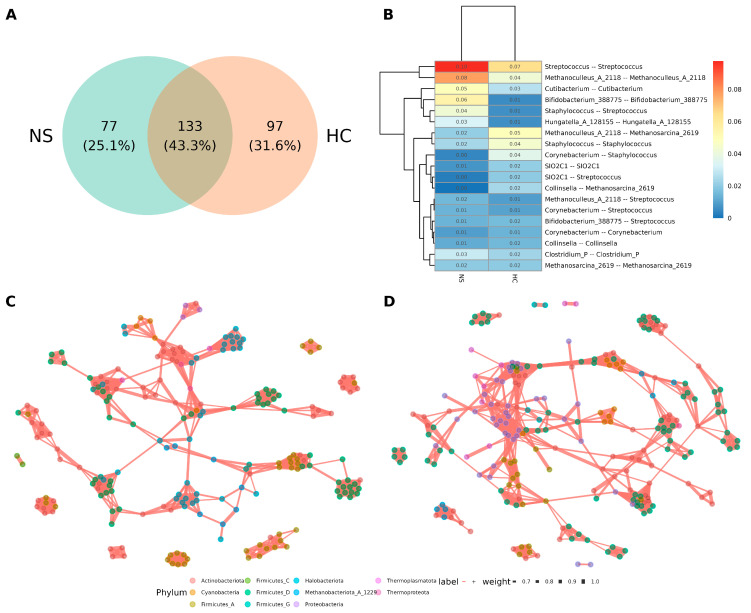
Microbial co-occurrence network comparison between human milk and neonatal stools. (**A**) Venn diagram of edges between the networks of neonatal stool (NS) and human colostrum (HC). (**B**) Number distribution of taxa associated with the linked nodes of positive edges in networks of NS and HC. The number in the plot indicates the ratio of edges against all the positive edges in the network. (**C**) Microbial co-occurrence network of neonatal stool. A connection between nodes stands for a strong (Spearman’s *ρ* > 0.6) and significant (*p* > 0.01) correlation. (**D**) Microbial co-occurrence network of human colostrum. A connection stands for a strong (Spearman’s *ρ* > 0.6) and significant (*p* > 0.01).

**Figure 6 microorganisms-13-00085-f006:**
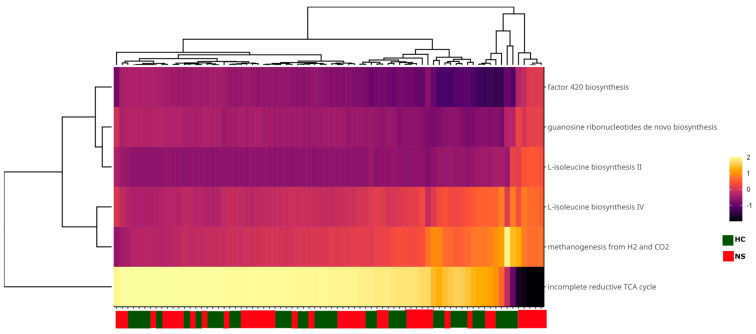
Heatmap of functional microbial metabolic pathways using PICRUSt2 analysis with MetaCyc database. Columns show the abundance of main metabolic pathways with a prevalence of at least 10% in the samples and an abundance > 1%. Sample names are shown on the *X*-axis. The color scale from black (−2) to white (2) indicates the relative abundance of the predicted metabolic pathways; the green and red color tags indicate the sample type.

**Table 1 microorganisms-13-00085-t001:** Sociodemographic and clinical characteristics of the study population.

Maternal Data		n (%)
	Age (years) ^a^	22.7 ± 6.7
	BMI ^b^	24.2 ± 4.22
	Birthplace	
	State-of-Mexico	29 (72.5)
	Mexico City	6 (15)
	Other (Puebla, Veracruz, etc.)	5 (12.5)
	Main Activity	
	Housewife	38 (95)
	Student	1 (2.5)
	General employee	1 (2.5)
	Educational level	
	Elementary school	20 (50)
	High school	18 (45)
	College	2 (5)
	Parity	
	Uniparous	18 (45)
	Multiparous	22 (55)
	Delivery mode	
	Vaginal	27 (67.5)
	C-Section (non-elective)	13 (32.5)
**Neonatal data**		**n (%)**
	Age at sample collection, days	
	<4	40 (100)
	Sex ^c^	
	Female	24 (60)
	Male	14 (35)

^a,b^ Expressed as mean ± standard deviation. ^c^, no available information for two samples. n-sample number, values in parenthesis represent the percentage.

## Data Availability

The datasets generated for this study can be found in the NCBI BioProject ID PRJNA1018680 Link https://www.ncbi.nlm.nih.gov/bioproject/PRJNA1018680 (accessed on 11 December 2023).

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
