# Peer review of "Human Milk Archaea Associated with Neonatal Gut Colonization and Its Co-Occurrence with Bacteria"

_microorganisms, 2025, doi:10.3390/microorganisms13010085_

Round 1

Reviewer 1 Report

Comments and Suggestions for Authors

In this article, 40 mother-neonate Mexican pairs were analyzed, regarding the archea component in breast milk and neonatal stools. BRILLIANT article. Exhaustive article.

-Only in References, there are 56 out of 109 (numerous references, and good ones) older than 2019, maybe some of them could be replaced with some more recent ones.

Author Response

Manuscript ID: microorganisms-3398902

Title: Human milk archaeal communities associated with neonatal gut colonization and their co-occurrence with bacteria.

Authors: Maricarmen Salas-López, Juan Manuel Vélez-Ixta, Diana Laura Rojas-Guerrero, Alberto Piña-Escobedo, José Manuel Hernández-Hernández, Martín Noé Rangel-Calvillo, Claudia Pérez-Cruz, Karina Corona-Cervantes, Carmen Josefina Juárez-Castelán, Jaime García-Mena

Reviewer 1 (Round 1)

Comments and Suggestions for Authors

Thank you for such an interesting article!

I must confess this is one of the best articles I reviewed for MDPI.

You used a lot of biochemistry data that are very useful in understanding the way that microbiota works. Also, the data regarding the genetics involved in microbiota helps the reader to have a more profound understanding of the gut microbiota.

You used a lot of graphical elements which is very supportive for the info provided by your article.

In my opinion the article can be published after a more thorough check of grammar and spelling.

Answer: We sincerely thank the reviewer for their positive feedback and thoughtful comments regarding the biochemical and genetic data, as well as the graphical elements in our article. We are especially grateful for your kind remarks on the quality and relevance of the content. In response to your suggestion, we have thoroughly reviewed the manuscript for grammar and spelling errors and made the necessary corrections to improve clarity and readability. We appreciate your careful review and constructive input, which have enhanced the quality of our work.

---end-of-text---

Reviewer 2 Report

Comments and Suggestions for Authors

Thank you for such an interesting article!

I must confess this is one of the best articles I reviewed for MDPI. 

You used a lot of biochemistry data that are very useful in understanding the way that microbiota works. Also, the data regarding the genetics involved in microbiota helps the reader to have a more profound understanding of the gut microbiota.

You used a lot of graphical elements  which is very supportive for the info provided by your article.

In my opinion the article can be published after a more thorough check of grammar and spelling.

Author Response

Manuscript ID: microorganisms-3398902

Title: Human milk archaeal communities associated with neonatal gut colonization and their co-occurrence with bacteria.

Authors: Maricarmen Salas-López, Juan Manuel Vélez-Ixta, Diana Laura Rojas-Guerrero, Alberto Piña-Escobedo, José Manuel Hernández-Hernández, Martín Noé Rangel-Calvillo, Claudia Pérez-Cruz, Karina Corona-Cervantes, Carmen Josefina Juárez-Castelán, Jaime García-Mena

Reviewer 2 (Round 1)

Comments and Suggestions for Authors

Human milk archaea associated with neonatal gut colonization and its co-occurrence with bacteria.

In this study, the authors aimed to characterize the archaeal composition in samples of mother-neonate pairs to observe the potential vertical transmission. A cross-sectional study was performed to characterize the archaeal diversity of human colostrum-neonatal stool samples by next-generation sequencing of V5-V6 16S rRNA gene libraries. Intra- and inter-sample analyses were carried out to describe the archaeal diversity in each sample type.

Answer: We sincerely thank the reviewer for their thoughtful and constructive comments.

Include P value and experimental design in the abstract

Answer: Regarding the suggestion to include the p-value and experimental design in the abstract, we respectfully believe that adding p-values might detract from the key findings, as their inclusion could lead to confusion without the full context of the analysis. The abstract aims to highlight the study’s main results, and detailed statistical results, such as p-values, are better discussed in the main body of the manuscript, where the analyses are presented in full.

L82-83: revise

Answer: the text mentioned in the referred lines was reviewed at the reviewer’s request.

Include a statement about the consent of human subjects who participated in this study.

Answer: Concerning the consent of participants, we would like to clarify that a statement regarding the consent process has already been included in lines 99-100, as per the guidelines for ethical considerations in human studies.

L204 and elsewhere: try to avoid the use of the word "we"

Answer: In response to the suggestion to reduce the use of "we," we have made revisions throughout the manuscript, including lines 204 and others, to minimize the use of first-person pronouns and maintain a more formal tone. Additionally, changes have been made to lines 82-83 to improve clarity. All changes are highlighted in yellow.

Materials and methods: well-explained

Figures: Clear and well-explained

A very good discussion

The conclusion is based on the findings of this study.

Answer: We also appreciate the positive feedback on the Materials and Methods, figures, discussion, and conclusion sections, and we are grateful for the reviewer’s time and suggestions, which have greatly contributed to the improvement of the manuscript.

---end-of-text---

Reviewer 3 Report

Comments and Suggestions for Authors

Human milk archaea associated with neonatal gut colonization and its co-occurrence with bacteria.

 In this study, the authors aimed to characterize the archaeal composition in samples of mother-neonate pairs to observe the potential vertical transmission. A cross-sectional study was performed to characterize the archaeal diversity of human colostrum-neonatal stool samples by next-generation sequencing of V5-V6 16S rRNA gene libraries. Intra- and inter-sample analyses were carried out to describe the archaeal diversity in each sample type.

Include P value and experimental design in the abstract

L82-83: revise

Include a statement about the consent of human subjects who participated in this study.

Materials and methods: well-explained

L204 and elsewhere: try to avoid the use of the word "we"

Figures: Clear and well-explained

A very good discussion

The conclusion is based on the findings of this study.

Author Response

Manuscript ID: microorganisms-3398902

Title: Human milk archaeal communities associated with neonatal gut colonization and their co-occurrence with bacteria.

Authors: Maricarmen Salas-López, Juan Manuel Vélez-Ixta, Diana Laura Rojas-Guerrero, Alberto Piña-Escobedo, José Manuel Hernández-Hernández, Martín Noé Rangel-Calvillo, Claudia Pérez-Cruz, Karina Corona-Cervantes, Carmen Josefina Juárez-Castelán, Jaime García-Mena

Reviewer 3 (Round 1)

Comments and Suggestions for Authors

In this article, 40 mother-neonate Mexican pairs were analyzed, regarding the archea component in breast milk and neonatal stools.

BRILLIANT article. Exhaustive article.

-Only in References, there are 56 out of 109 (numerous references, and good ones) older than 2019, maybe some of them could be replaced with some more recent ones.

Figures: Clear and well-explained

A very good discussion

The conclusion is based on the findings of this study.

Answer: we thank you for the encouraging remarks regarding the work done by the reviewer. We have carefully considered the observation about the references in number and dates. We believe the referred works are the appropriate reports to support the scientific statements.

---end-of-text---
